# Deep audio embeddings for vocalisation clustering

**Paul Best** ⬤ *, Sébastien Paris, Hervé Glotin, Ricard Marxer ⬤

Université de Toulon, Aix Marseille Univ, CNRS, LIS, Toulon, France

* paul.best@univ-tln.fr

**Data Availability Statement:** The Python package to run repertoire discovery procedures, as well as codes used in this study, are available at https://gitlab.lis-lab.fr/paul.best/repertoire_embedder/. Auto-encoder weights, vocalization embeddings, and ground truth labels of vocalization types that are necessary to produce performance metrics and

## Abstract

The study of non-human animals' communication systems generally relies on the transcription of vocal sequences using a finite set of discrete units. This set is referred to as a vocal repertoire, which is specific to a species or a sub-group of a species. When conducted by human experts, the formal description of vocal repertoires can be laborious and/or biased. This motivates computerised assistance for this procedure, for which machine learning algorithms represent a good opportunity. Unsupervised clustering algorithms are suited for grouping close points together, provided a relevant representation. This paper therefore studies a new method for encoding vocalisations, allowing for automatic clustering to alleviate vocal repertoire characterisation. Borrowing from deep representation learning, we use a convolutional auto-encoder network to learn an abstract representation of vocalisations. We report on the quality of the learnt representation, as well as of state of the art methods, by quantifying their agreement with expert labelled vocalisation types from 8 datasets of other studies across 6 species (birds and marine mammals). With this benchmark, we demonstrate that using auto-encoders improves the relevance of vocalisation representation which serves repertoire characterisation using a very limited number of settings. We also publish a Python package for the bioacoustic community to train their own vocalisation auto-encoders or use a pretrained encoder to browse vocal repertoires and ease unit wise annotation.

## Introduction

### Context

Many animal species use acoustics to communicate. Often in this context, signals take the form of distinct units (or vocalisations) arranged in sequences, with information potentially being carried through order and/or rhythm [1]. Like the discrete communication channels formalised by Shannon [2], the elements of the sequences often seem to emerge from a finite set of discrete categories. This finite set of possible units is referred to as the repertoire of a communication system. Vocal repertoires can be specific to groups of individuals (*e.g.*, orca *Orcinus orca* [3]), populations (*e.g.*, beluga whale *Delphinapterus leucas* [4] or mountain white-crowned sparrows *Zonotrichia leucophrys oriantha* [5]), or whole species (*e.g.*, common

plots in this study are available on a figshare repository: https://doi.org/10.6084/m9.figshare.23138210.v1 The acoustic data that were used to train auto-encoders and generate embeddings along with their expert label are not owned by the authors but might be accessed via their respective sources: bengalese finch1: Nicholson, David; Queen, Jonah E.; J. Sober, Samuel (2017): Bengalese Finch song repository. figshare. Dataset. https://doi.org/10.6084/m9.figshare.4805749.v5 bengalese finch2: Koumura, Takuya (2016): BirdsongRecognition. figshare. Media. https://doi.org/10.6084/m9.figshare.3470165.v1 cassin vireo, california thrasher, black-headed grosbeak: Arriaga, J. G., Cody, M. L., Vallejo, E. E., & Taylor, C. E. (2015). Bird-DB: A database for annotated bird song sequences. Ecological Informatics, 27, 21-25. https://taylor0.biology.ucla.edu/birdDBQuery/ humpback whale and humpback whale (small) (upon request): Malige, F., Djokic, D., Patris, J., Sousa-Lima, R., & Glotin, H. (2021). Use of recurrence plots for identification and extraction of patterns in humpback whale song recordings. Bioacoustics, 30(6), 680-695. (contact: Franck Malige, franck.malige@lis-lab.fr) Dolphin (upon request): Sayigh L, Janik VM, Jensen F, Scott MD, Tyack PL, Wells R. The Sarasota dolphin whistle database: A unique long-term resource for understanding dolphin communication. Frontiers in Marine Science. 2022;. (contact: Laela S. Sayigh, lsayigh@whoi.edu).

**Funding:** Hervé Glotin received the grants ANR-20-CHIA-0014 and ANR-21-CE04-0019, and Ricard Marxer received the grant ANR-20-CE23-0012-01 from Agence Nationale de la Recherche. These financed salaries and computers to run experiments. The funders had no role in study design, data collection and analysis, decision to publish, or preparation of the manuscript. To collect the humpback whale data, Renata Sousa-Lima received grants from Fundação O Boticário de Proteção à Natureza / MacArthur Foundation and the Society for Marine Mammalogy (Small-Grants-in-Aid of Research). The funders had no role in study design, data collection and analysis, decision to publish, or preparation of the manuscript.

**Competing interests:** The authors have declared that no competing interests exist.

marmosets *Callithrix jacchus* [6] or swamp sparrows *Melospiza georgiana* [7]). They can be learnt or genetically determined [8], and they are central to the study of communication systems.

Indeed, bioacousticians often encode acoustic signals as sequences of tokens, each associated with categories of a repertoire [1]. Vocal sequence transcription facilitates processes such as sequence comparison or pattern extraction, especially as compared to dealing with raw data (*i.e.*, such as time series of sound pressure levels). This task is analogous to speech transcription, with the crucial difference that the operator does not comprehend the signal it transcribes.

This process, when conducted on non-human animals, is subject to debate for the human biases it might introduce. A human's categorisation of vocalisations might not capture the essential complexity of a communication system [9] and arbitrary decisions might occur regarding group granularity [10], all potentially leading to inter-annotator disagreements [11]. Nonetheless, to this day, human annotators perform better than machines at classifying vocalisations [12, 13], and human categorisations have widely been validated with observations on the emitter(s) (*i.e.*, at the individual level [13] or the group level [3]) or on the call's function [14].

Regardless of accuracy, characterising non-human vocal repertoires manually can be a very laborious task. A thorough examination of large vocal sequences is needed to decide on repertoire categories (stereotype and boundaries), and when dealing with large repertoires (up to dozens of categories), it can be very demanding if not impossible to consider the totality simultaneously and define categories in a systematic way.

The transcription of vocal signals matters for it allows unveiling structure [15] (syntax), meaning [14] (semantics), social structures [3, 16] and cultural patterns [17] in non-human animal communication systems. The manual annotation of vocalisations by type is laborious and could be biased whereas fully unsupervised methods are not yet reliable enough. Submitting clusters of similar vocalisations to a human operator could alleviate these limitations and contribute to advances in animal communication studies. This paper thus intends to explore and compare vocalisation representations for the clusters they yield, leading to computer assisted vocal repertoire discovery procedures.

## Vocalisations feature extraction and clustering

Automatic clustering of animal vocal repertoires has been studied in the past. Methods vary but usually revolve around three main steps: feature extraction of signals, dimensionality reduction, and clustering (Fig 1).

To represent acoustic signals, one can make use of Predefined Acoustic Features (PAFs), *i.e.*, handcrafted temporal and spectral signal characteristics. For instance, depending on the signals to be clustered, features can describe the temporal envelope, the frequency contour, or the response to filterbanks (*e.g.*, Mel or gammatone). Researchers have used PAFs to cluster vocalisations of zebra finches (*Taeniopygia guttata*) [18], baboons (*Papio ursinus*) [19], bottlenose dolphins (*Tursiops truncatus*) [20], gibbons (*Hylobates funereus*) [21], and mice (*Mus musculus*) [22, 23]. For instance, Elie et al. [18] used 22 PAFs extracted using the Biosound package [24] to cluster zebra finch (*Taeniopygia guttata*) vocalisations, Sainburg et al. [25] used 18 features from the same package to visualise and cluster vocalisations from 20 species, Clink and Klinck [21] used Mel Frequency Cepstral Coefficients (MFCCs) to cluster gibbon (*Hylobates funereus*) calls by individual, and Van Segbroeck et al. [22] used a gammatone filterbank to cluster mice (*Mus musculus*) vocalisations. Alternatively, to capture spectro-temporal variations, the concatenation of consecutive spectrogram frames can be used [25, 26].

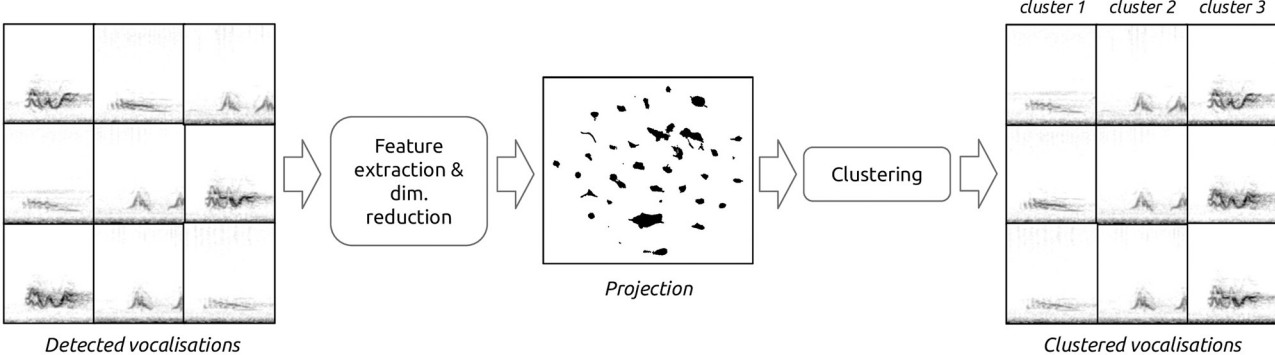

**Fig 1. Main steps involved in vocal repertoire clustering.** This paper focuses mainly on the feature extraction step.

Often, we do not know a priori which feature will be the most discriminant for vocalisation types, and high dimensional spaces make similarity measurements difficult (curse of dimensionality). This motivates using dimensionality reduction algorithms such as Principal Component Analysis (PCA) [18, 20, 25] or Uniform Manifold Aproximation (UMAP) [25–27] to emphasise the most 'relevant' features for a given dataset. One important limitation of using PAFs is that there is no generic set that suits all vocal repertoires, and some of the features need specific settings depending on vocalisation frequency ranges or signal to noise ratio (SNR). An opportunity to avoid having to manually choose the right set of features and tune their settings is to use features learnt using deep representation learning (as opposed to hand-crafted features like PAFs). Following this approach, auto-encoder artificial neural networks have been used by Goffinet et al. [28] on mice and zebra finch vocalisations, by Bergler et al. [29] to cluster orca calls, by Rowe et al. [30] to cluster bird vocalisations by species, and by Tolkova et al. [31] to discriminate between background noise and bird vocalisations.

Auto-encoders are artificial neural networks self-supervisedly trained (without the need for labels) to encode data into a lower dimensional space (called bottleneck). To ensure the conservation of the information in the bottleneck, the encoder network is optimised jointly with a decoder network to maximise the resemblance between the decoded encoded data and the input data (reconstruction loss).

Once we obtain a high level representation of vocalisations, whether with PAFs or with auto-encoder embeddings, and whether dimensionaly reduced or not, clustering algorithms allow to group close points (in the high-level feature space) together to form discrete classes. In the literature of vocal repertoire clustering, chosen algorithms were mostly K-Means and HDBSCAN [32], HDBSCAN usually used after dimensionality reduction [25, 26].

To follow up on this research, we propose studying vocalisation clustering, especially the choice of audio representation and its effect on the subsequent clustering. Auto-encoders trained on vocalisation spectrograms have recently shown promising results [28, 29], we thus compare the embedding space they yield to that of more traditional methods, especially how it allows to cluster units by acoustic similarity. Taking detected vocalisations as an input, clusters of similar vocalisations can thus be presented to an operator, facilitating vocal repertoire characterisation as well as annotation (*e.g.*, to train a classifier model). To quantify the performance of the varying methods, we use them on 8 datasets of animal vocal repertoires (including birds and cetaceans) and measure their agreement with expert labels of vocalisation types. Along with the performance analysis of the system across varying hyper-parameters, we publish a python interface for biologists to use deep audio embeddings for their own vocal repertoire discovery procedures.

## Materials and methods

### Datasets

In this study, we use 8 different datasets that gather large amounts of timestamped and type labelled vocalisations. These comprise 4 birds and 2 marine mammals species. In most cases, unit classes are shared across recorded individuals, describing a population or species specific repertoire. The exceptions are the bengalese finch datasets which contain individual specific repertoires and the bottlenose dolphin dataset which contains signature whistles [33] (signature whistles are not a vocal repertoire but still inform on the relevance of representations for vocalisation categorisation). In these cases, classes are defined as the combination of unit type and individual identity. Table 1 summarises the characteristics of each dataset.

In the study by Malige et al. [37] for which humpback whale vocalisations were originally annotated, around 7,500 vocalisations were labelled by an expert, and to assess inter-annotator disagreements, a second analyst also annotated a sub-set of 1,800 vocalisations, all belonging to recordings of a single day. Both sets of labels are used in this study separately, with the latter referred to as 'humpback whale (small)'.

For some datasets, not all vocalisations were labelled by unit type. Moreover, this paper studies fully unsupervised methods: labels only serve the final evaluation of the yielded clustering. Therefore, to stay close to a realistic scenario of having a potentially noisy set of detected vocalisations, all detections were used in the procedures preceding the evaluation. Additionally, in some cases, classes were highly underrepresented (less than 20 occurrences) or seemed to result from labelling errors. These vocalisations were considered as unlabelled.

### Signal preprocessing and Fourier transform

For the procedure described in this paper, the only parameters that are dataset specific are related to low-level signal preprocessing, and require only a limited amount of knowledge about the repertoire of interest to be set properly. They are the sample duration $\mathcal{T}$, the sampling rate fs and the Fast Fourier Transform (FFT) window size NFFT, which Table 2 reports for each dataset. Note that the maximum frequency for Mel filterbanks was always set to the nyquist frequency, and so is defined by the sampling rate implicitly. This section describes how these parameters were set, along with the full procedure that compiles signals into spectrograms to feed to the auto-encoder.

A consideration for setting spectrogram parameters is that auto-encoder's decoders typically reconstruct images by successive factor two up-sampling. Input spectrograms which will

**Table 1. Size of each dataset used in the experiments.**

| Species and source | # Units | # Vocalisations | % Labelling |
|---|---|---|---|
| bengalese finch [34] | 33 | 179,864 | 99 |
| bengalese finch [35] | 93 | 215,037 | 100 |
| california thrasher [36] | 12 | 33,300 | 4 |
| cassin vireo [36] | 102 | 144,185 | 46 |
| black-headed grosbeak [36] | 37 | 35,574 | 18 |
| humpback whale [37] | 15 | 7,495 | 98 |
| humpback whale (small) [37] | 12 | 1,800 | 100 |
| bottlenose dolphin [33] | 20 | 400 | 100 |

Dataset characteristics include the number of different types, the total number of detected vocalisations, and the proportion of vocalisations that are labelled by type.

**Table 2. Dataset specific spectrogram settings.**

| Species and source | fs (kHz) | NFFT | Hop (ms) | $\mathcal{T}$ (s) |
|---|---|---|---|---|
| bengalese finch [34] | 32 | 256 | 0.7 | 0.1 |
| bengalese finch [35] | 32 | 256 | 0.7 | 0.1 |
| california thrasher [36] | 44.1 | 512 | 1.8 | 0.25 |
| cassin vireo [36] | 44.1 | 512 | 3.8 | 0.5 |
| black-headed grosbeak [36] | 44.1 | 512 | 2.6 | 0.35 |
| humpback whale [37] | 11.025 | 1024 | 14.9 | 2 |
| humpback whale (small) [37] | 11.025 | 1024 | 14.9 | 2 |
| bottlenose dolphin [33] | 96 | 512 | 15.6 | 2 |

be matched with auto-encoder reconstructions thus need to be of dimensions that follow ($k_f 2^n$ $\times k_t 2^n$), with $k_f$ and $k_t$ integers and $n$ the number of factor two up-sampling (in the proposed architecture $n = 5$, more details are given in the next section). For all datasets in this study, 128 frequency bins and 128 temporal bins appeared to suffice in containing vocalisation details; all spectrograms were thus set to be of size 128x128 ($k_f = k_t = 4$). Note however that if more spectral or temporal bins were needed for a new species, the same auto-encoder architecture could allow to manage 256 bins or more.

Preprocessing first consists in extracting the signal surrounding the center of the annotation. The sample duration $\mathcal{T}$ was fixed for each dataset. In order to fully contain most vocalisations while avoiding hiding details for small ones, it was set at the 3rd quartile of all vocalisation durations. In the case of pre-cut vocalisation files with a smaller duration than this fixed value, signals were zero-padded. The sampling rate was fixed to the most common value in the dataset, and vocalisations with a different sampling rate were resampled to match the common value using the scipy python package [38]. The exception is the humpback whale dataset, for which the sampling rate was reduced to 11,025 Hz in order to reduce the nyquist frequency and increase the frequency resolution for the relatively low frequency vocalisations. Resulting signals are then z-normalised before its frequency decomposition.

The choice of Fourier transform parameters define the spectro-temporal resolution of spectrograms, and need to suit spectro-temporal modulation rates of vocalisations. For this study, FFT window sizes were manually set by quick spectrogram inspections or by borrowing from studies publishing the databases. For the short term Fourier transform, we used unpadded Hann windows and a hop size set to yield spectrograms of 128 time bins (Hop $= (\mathcal{T} \times \text{fs} - \text{NFFT}) \times \frac{1}{128}$). The Fourier transform comes with the limitation of having to choose a fixed spectro-temporal resolution, which wavelet based transforms can help alleviate [39], allowing a better representation especially for transient signals. However, choosing such approach would imply a significant increase in computation, and additional dataset specific settings (e.g., choice of wavelet family or hyperparameters). For the sake of simplicity and usability, and as the studied signals are not transient, we chose the general purpose Fourier transform in our experiments.

Following the short term Fourier transform, several frequency and dynamic range compressions were tested. Mel filterbanks with 128 filters between 0 and the nyquist frequency were used, and compared to keeping the spectrogram with a linear frequency layout. In the latter case, maximum pooling was used to reduce the number of frequency bins down to 128 and match other spectrograms, independently of the FFT size. As for dynamic range compression of spectro-temporal magnitudes, we compared no compression against logarithmic and PCEN

[40] compression. In all cases, spectro-temporal magnitudes were z-normalised before applying the auto-encoder.

## Auto-encoder network architecture

Contrary to supervised classifier neural networks for which the bioacoustics community often uses off-the-shelf architecture [41], there is to our knowledge not yet an architecture of choice for auto-encoder networks. Bergler et al. [29] use a ResNet-18 architecture, and Goffinet et al. [28] use a custom architecture consisting in successive blocks of convolution, batch normalisation and rectifier linear units (ReLU). For the encoder, these successive blocks gradually increase the number of feature maps while decreasing the spectro-temporal dimensions. This is a common practice in deep convolutional networks, following the intuition of deeper layers having a higher level representation of the data and a larger receptive field. Symmetrically, the decoder network decreases the depth and increases spectro-temporal dimensions.

We followed a similar approach, with our proposed encoder consisting in a succession of 5 convolutional blocks, all with kernels of size 3 by 3 (Fig 2). To increase the depth of the data, the number of kernels starts at 32 and is multiplied by 2 at each block until the last. To decrease the spectro-temporal dimensions, convolutions operate with a stride of 2. The output of each convolution except the last are batch normalised before ReLUs are applied.

The decoder network is also composed of 5 blocks, this time each applying factor 2 up-sampling followed by 2 successive convolution blocks (convolution with kernels of size 3, batch-normalisation and ReLU). The rationale for having two convolution blocks after each up-sampling is to have a larger receptive field than with a single one.

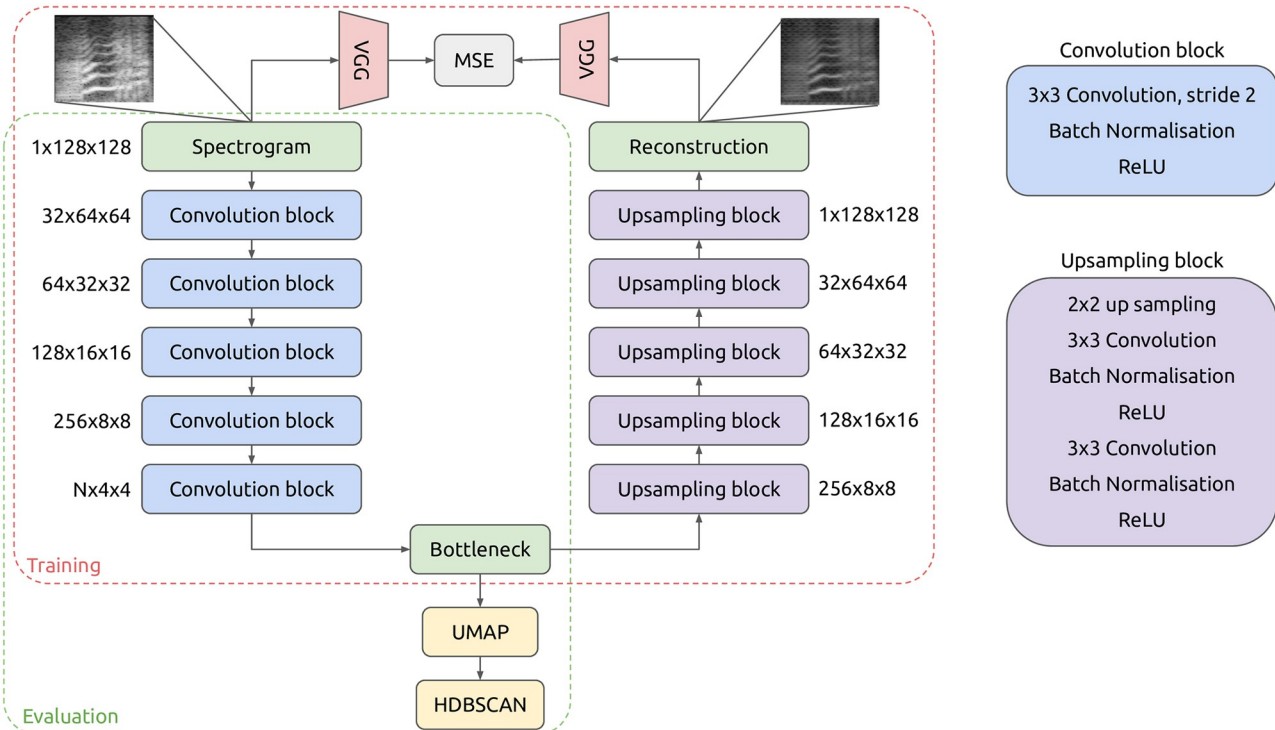

**Fig 2. Architecture of the auto-encoder.** During training, VGG embedings are used to compare the reconstruction with the input (perceptual loss). For evaluation, only the encoder is used, followed by dimensionality reduction and clustering.

## Training procedure

When training an auto-encoder, the main constraint that we impose on the network is that the information contained in the input is maintained in the bottleneck. To do so, we ensure that, from the bottleneck, the decoder network is able to reconstruct something similar to the original input. A straightforward approach to quantify this similarity is to compute the Mean Square Error (MSE) between the input and the reconstruction, *i.e.*, here the average of the squared difference for each spectro-temporal bin. However, for our use case, we mostly want the frequency contour of vocalisations to be well reconstructed, which might represent only a small proportion of the spectro-temporal bins. To have a comparison that emphasizes more on foreground content than on background noise, we can use the perceptual loss [42], in which we compute the MSE between embeddings of a third party pre-trained network (Fig 2). Here we make use of the VGG model trained on ImageNet for image classification [43].

Auto-encoders were trained on each dataset independently, with an Adam optimiser [44] using batches of 128 vocalisations. The training was stopped when the median of the loss for the last thousand steps did not decrease as compared to the previous thousand steps.

## Evaluation

Once the auto-encoder is trained, we can use the bottleneck embedding to represent vocalisations. The intuition is that this relatively compact representation (spectrograms were initially encoded in 128x128 dimensions) will help in measuring vocalisation similarity, and thus will be correlated with expert unit classification. In other words, close points in this embedding space would be perceived as similar by an expert and categorised as from the same type.

To verify this intuition and measure the agreement between embedding similarity and experts' perceptual similarity, we cluster close points in the embedding space and compare the result with expert unit classification. To mitigate the curse of dimensionality which can strongly hinder clustering performance, we reduce the dimensionality of our data using the UMAP algorithm [27].

In the resulting embedding space (Fig 3), we can now group vocalisations that are close together with a systematic approach, and compare the resulting clusters with expert classes. Having no prior constraint on linear separability or scale, density based clustering seems

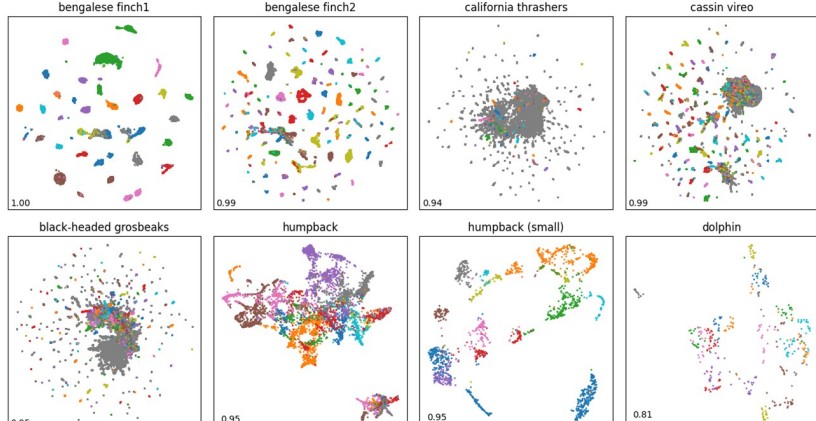

**Fig 3. UMAP projection of AE embedding for each dataset.** Points are colored by expert labels (not clusters), or in grey when no annotation is available. The Hopkins statistic is displayed at the bottom left of each scatter plot.

appropriate for these distributions of points. We thus chose the HDBSCAN algorithm [45] using its public python implementation [32].

To evaluate the agreement between clusters that emerges from the learnt representation and expert labels, we use the Normalised Mutual Information (NMI). It is a common metric for clustering performance evaluation which is also equivalent to the V-measure that is found in other vocal repertoire discovery studies [25]. Given a set of clusters $C$ and a set of labels $L$, the NMI measures the relative entropy between the product of marginal distributions $P_L \times P_C$ and the joint distribution $P_{L,C}$, and normalises it so that a perfect match gives 1 and fully unrelated categorisations give 0. Eq 1 formulates the NMI, with H denoting the entropy and $D_{KL}$ the Kullback–Leibler divergence or relative entropy.

$$\mathrm{NMI}(L; C) = \frac{D_{KL}(P_{L,C} \| P_L \otimes P_C) \times 2}{\mathrm{H}(L) + \mathrm{H}(C)}, \tag{1}$$

## Vocal repertoire discreteness

The learnt representation, besides its potential agreement with experts' perception of similarity for categorisation, can help us learn the discreteness of vocal repertoires which varies across animal species [19]. Indeed, the clusterability of a dataset can be measured using the Hopkins statistic [46] (Eq 2), which we can apply to vocalisations' latent distributions [25].

$$\mathrm{Hopkins} = \frac{\sum_{i=1}^{m} u_i}{\sum_{i=1}^{m} u_i + \sum_{i=1}^{m} w_i}. \tag{2}$$

This measure compares the sum of distances $u_i$ between $m$ points sampled from a random distribution and their closest neighbour in a dataset, with the sum of distances $w_i$ between $m$ points sampled from the dataset and their closest neighbour. Here, $m$ was set to 100 and a normal distribution was estimated using the dataset's moments. The Hopkins statistic computed on UMAP embeddings is reported for each dataset in Fig 3.

## Handcrafted features and deep audio embeddings

To have a baseline of comparison for the clustering performances of learnt features against handcrafted ones, we ran experiments using two feature sets commonly found in the litterature, *i.e.*, whole spectrograms and PAFs. Following Sainburg et al. [25], we computed 32x32 spectrograms for each vocalisation and used the magnitudes of each spectro-temporal bin as independent features. For the PAFs, we used the set of 18 features used in the latter study, extracted using the Biosound package [24]. To allow the comparison between these feature extraction methods and the auto-encoder's, for all experiments, sample durations, sampling rates and Fourier window size remained unchanged.

Additionally, to get a sense of how relevant is the auto-encoder learning framework, we projected vocalisations using third party neural networks of the Holistic Evaluation of Audio Representations (HEAR) [47]. We used the 3 baseline models of this challenge that give relatively generic audio embeddings and were evaluated on a wide range of downstream tasks including speech, music, and environmental sound analysis. These models include Wav2Vec2 [48] (trained on speech), CREPE [49] (trained on synthesised music), and OpenL3 [50] (trained on audio / video content correspondance).

Focusing here on the feature extraction step, the dimensionality reduction (UMAP) and clustering (HDBSCAN) were kept identical as for the auto-encoder evaluation procedure.

## Results

### Hints on procedure's optional settings

The focus of this paper is to study the impact of choosing an auto-encoder based representation for vocalisation clustering. However, to evaluate this approach and compare performances with other feature extraction methods, we rely on dimensionality reduction and clustering algorithms. The choice of algorithm (here UMAP and HDBSCAN respectively) along with their hyper-parameters can have a strong impact on performances. Additionally, settings such as spectrogram dynamic range compression, frequency layout, or the auto-encoder's bottleneck size also might affect the learnt representation and by extension clusters' agreement with expert categorisation. Rather than reporting extensively on performance variations in relation to this large search space, the main results of this study rely on a fixed set of settings. This section presents them along with the rationales and empirical findings behind the different choices that were made.

**Spectrogram configuration.**  As for spectrogram magnitude normalisation and frequency layouts, no configuration performs better in all cases. This is due to the diversity of frequency ranges and signal to noise ratio found across datasets. The Mel frequency layout appears mostly beneficial, but the effect of dynamic range compression (whether none, PCEN, or logarithmic) is dataset specific (S1 Fig). For its good compromise across datasets and species, we suggest using the Mel frequency layout with logarithmic compression as a first choice, and selected it for the following experiments.

**Dimensionality reduction.**  An early intuition of this study was that lower dimensional bottlenecks would lead to the elimination of unnecessary information (*e.g.*, small vocalisation variations or background noise) and thus yield better clustering results. Experiments with varying bottleneck sizes (from 16 to 512) showed a relatively low impact on the system's final performances (variations below 5%) with a minor tendency for higher performances with larger bottlenecks. This observation might be explained by a loss of information when using small bottlenecks, and the fact that UMAP is sufficient to reduce the dimensionality while preserving local distances. We thus set a bottleneck size of 256 for the following experiments.

Dimensionality reduction such as UMAP is useful for data visualisation (Fig 3), but can also help in clustering applications [25, 26]. In the latter case, there is no constraint for the embedding space to be bi-dimensional, making its number of dimension another setting that could affect the quality of the following clustering. By compressing the data, we reduce the curse of dimensionality for distance estimations and we potentially lose information, which is desirable to some extent. We experimented with UMAP compression in 2, 4, 8, 16 and 32 dimensions, and found that this setting has a relatively low impact on the NMI (mostly around 2% of variation), except for some cases like using only 2 dimensions for spectrogram representations (Fig 4). We thus suggest using more than 2 dimensions for clustering applications, unless an interactive browsing of embeddings is desired in which case bi-dimensional embeddings should not be too detrimental if using auto-encoder based representations. As for the following experiments, 8-dimensional embeddings were used.

**HDBSCAN configuration.**  Before going for the HDBSCAN algorithms, trials were conducted by applying K-means directly on auto-encoder embeddings. Performances always staid below those of density based clustering after dimensionality reduction, which motivated to focus on the latter for following experiments. For the HDBSCAN clustering algorithm [32], the user can specify several settings: the minimum number of points for a dense region to be considered a cluster (minimum cluster size); the minimum amount of neighboring points to form a local dense region (minimum samples); an espilon value to merge clusters that are close together [51]; and the cluster selection algorithm which chooses whether to split large clusters

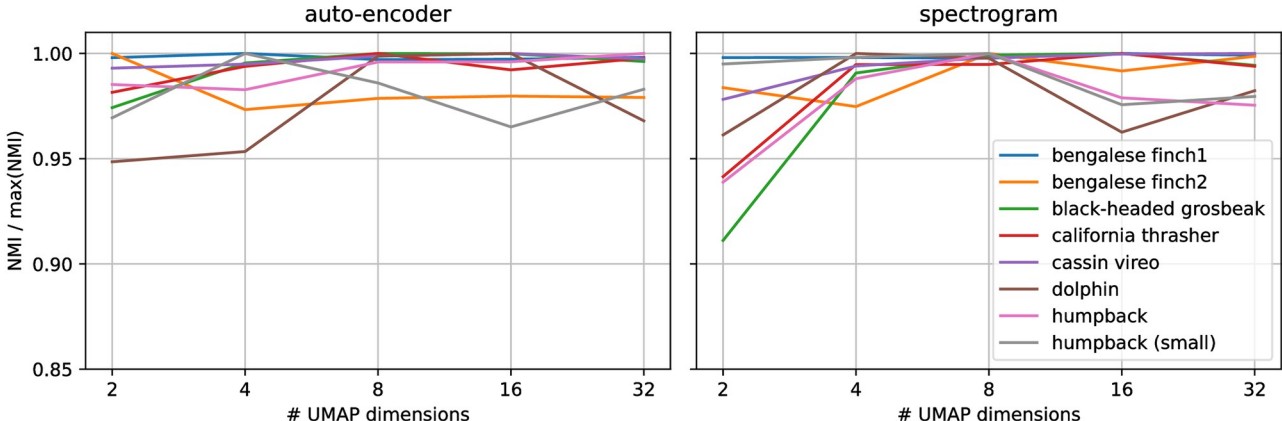

**Fig 4. Effect of varying UMAP compression levels on the subsequent categorisation.** Scores are reported as a ratio of the measured NMI divided by its maximum value across all compression levels, for the auto-encoder based representations (left) and the spectrogram representations (right).

into sub-units or not. The minimum cluster size can be approximated using prior information such as the total number of vocalisations and the expected number of unit types. Besides, the examination of vocalisation embeddings can help in getting a sense of dense regions, giving indications on how to set the three remaining parameters. We used a grid search to systematically tune these parameters for all datasets and found that the best compromise in performances is for a minimum cluster size of 10; minimum samples of 3; epsilon of 0.1; and the leaf cluster selection algorithm, except for the humpback whale datasets for which the EOM algorithm is beneficial. This exception aside, across all datasets, these 'generic' settings do not reduce the NMI by more than 4% as compared to optimal ones, and thus were kept for the following experiments. For experimenting with new species, we suggest to start of with these parameters and to inspect clustered embeddings in case some tuning is necessary.

## Comparison with other representations

**Handcrafted features.** Fig 5 summarises the NMI between clusters and expert labels for each dataset, comparing handcraft feature extractions with the auto-encoder method. The first

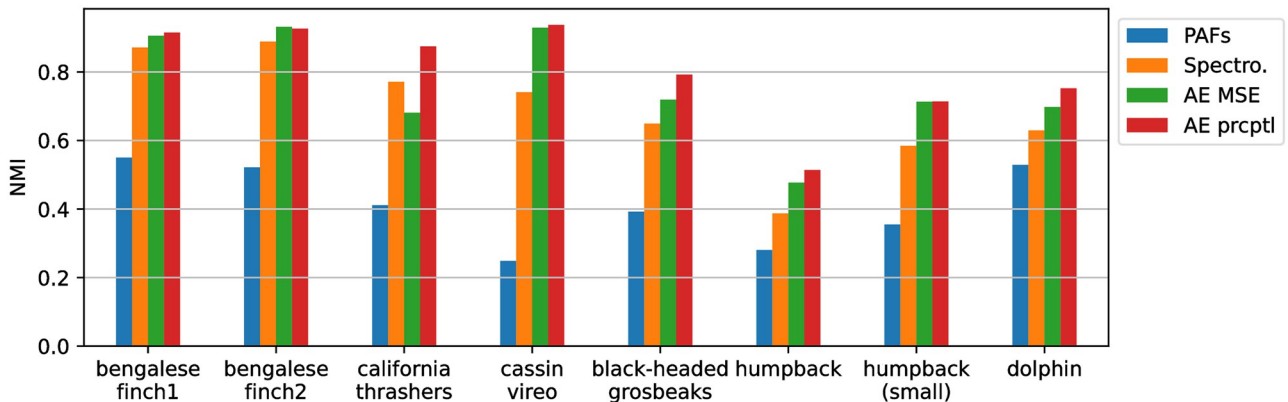

**Fig 5. Agreement between found clusters and expert labels depending on vocalisation representations.** Scores for the auto-encoder (AE) are given with and without the perceptual loss.

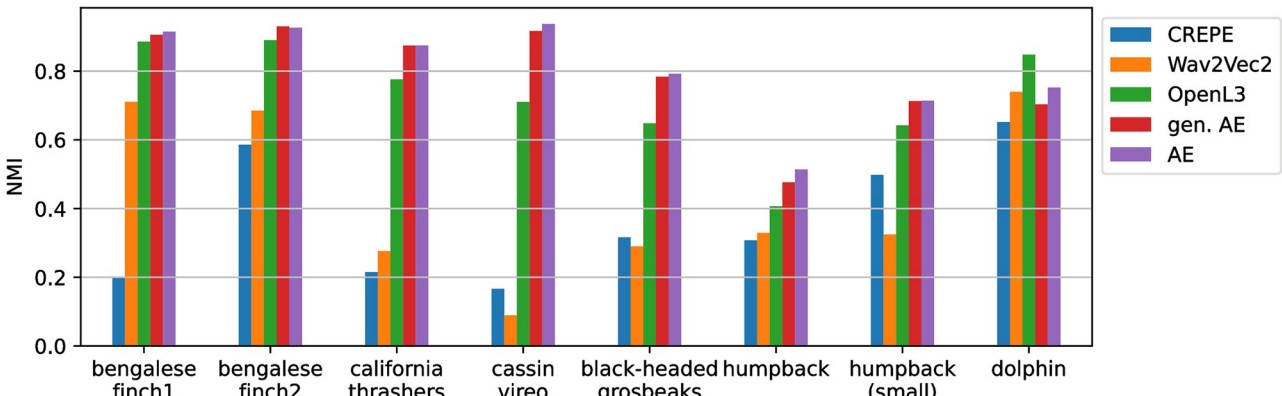

**Fig 6. Agreement between found clusters and expert labels for varying deep audio embeddings.** Scores are given for embeddings of the auto-encoder trained on the dataset it is tested on (AE), for auto-encoders trained on all other datasets (gen. AE), and for models of the HEAR challenge baseline.

insight that Fig 5 reveals is that learning features using an auto-encoder will yield better clustering results (in terms of agreement with expert labels) than using PAFs or whole spectrograms. Also, training the network with a perceptual loss instead of a regular pixel loss is almost systematically beneficial.

**Deep audio embeddings.** Deep learning models sometimes yield embeddings that are useful for tasks that they were not specifically trained for. For instance, the OpenL3 model trained on audio / video correspondence from youtube content can inform on the presence of a queen in a beehive [47]. Fig 6 reports on the relevance of such audio embeddings for the vocalisation clustering task studied here. Among the three models of the HEAR baseline, OpenL3 model gives the best overall performances. Nonetheless, with the exception of the dolphin whistle dataset, the auto-encoder gives better results than all third party task models.

In addition to the HEAR challenge baseline models, we give the performance of auto-encoders when trained on all datasets except the one it is tested on. Doing so, we test the potential for a generic vocalisation encoder that could inform on their similarity without needing any dataset specific training (note that since the humpback whale datasets share the acoustic signal they were treated as one in this experiment). Results in Fig 6 suggest a good potential for such 'generic' encoder, with performances being very close to that of dataset specific auto-encoders.

## Interpretable metrics for repertoire annotation

The most common use case of the procedure studied here is when biologists want to cluster similar vocalisations together, for instance for vocal sequence transcription. Using pre-clustered vocalisations (Fig 7) can drastically reduce the amount of effort necessary to sort out vocalisations, especially if the vocal repertoire is large or has not yet been catalogued. In this sense, the NMI is hardly interpretable in terms of how much time it saves an analyst. To mitigate this shortcoming, Table 3 reports on metrics that give insights on the amount of annotation effort spared / still needed after clustering was conducted, and so in the case of auto-encoder based representations and spectrogram representations.

The first step of an annotation process following a clustering algorithm is to go through the clusters looking for discriminant ones. A discriminant cluster is a cluster that gathers mostly similar vocalisations that would be considered as from the same type. Table 3 thus reports on

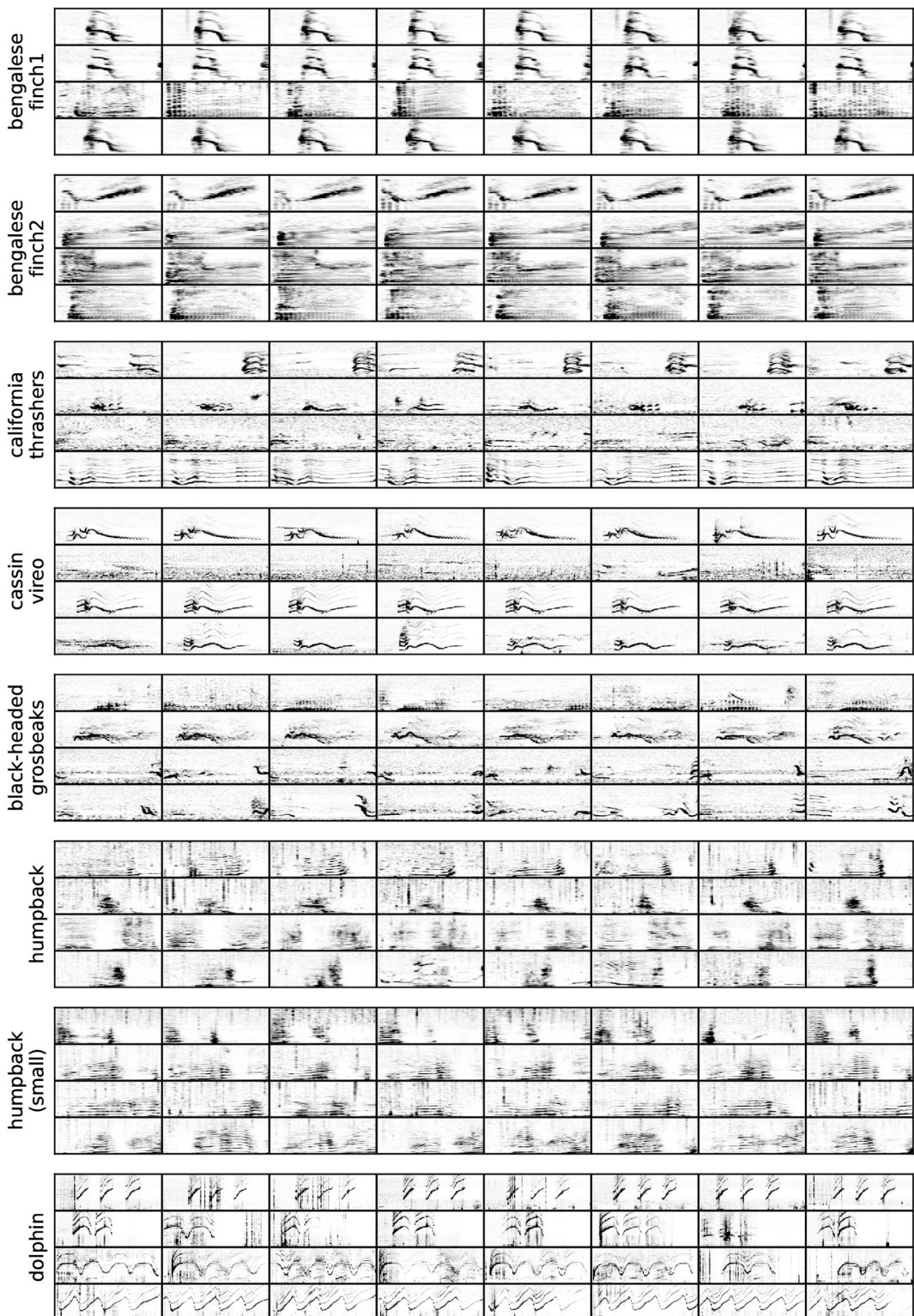

**Fig 7. Vocalisations as clustered by the proposed algorithm.** For each dataset, 4 clusters were randomly sampled (one per line) from which 8 vocalisations were in turn randomly sampled and plotted.

**Table 3. Results of the clustering procedure.**

| Species and source | # labels | # clusters | | % discr. | | % clustered | | # missed | |
|---|---|---|---|---|---|---|---|---|---|
| | | AE | Spec | AE | Spec | AE | Spec | AE | Spec |
| bengalese finch [34] | 33 | 61 | 94 | 95 | 92 | 93 | 89 | 3 | 3 |
| bengalese finch [35] | 93 | 165 | 167 | 96 | 92 | 93 | 94 | 0 | 3 |
| california thrasher [36] | 12 | 671 | 720 | 97 | 90 | 16 | 20 | 0 | 0 |
| cassin vireo [36] | 102 | 585 | 771 | 79 | 67 | 47 | 45 | 2 | 4 |
| black-headed grosbeak [36] | 37 | 702 | 748 | 81 | 64 | 43 | 27 | 1 | 1 |
| humpback whale [37] | 15 | 116 | 153 | 46 | 51 | 51 | 29 | 1 | 3 |
| humpback whale (small) [37] | 12 | 30 | 37 | 65 | 45 | 52 | 35 | 4 | 3 |
| bottlenose dolphin [33] | 20 | 16 | 12 | 47 | 31 | 32 | 15 | 12 | 16 |

Metrics include the total number of clusters, the proportion of discriminant clusters, the proportion of vocalisations that belong to the latter, and the amount of labels that are not included in the latter). Each are given for the auto-encoder based representation (left) and the spectrogram based representation (right).

the total number of clusters that the expert can browse into, as well as the proportion of clusters that are discriminant. Here, we consider a cluster to be discriminant if at least 90% of the labelled vocalisations it contains are from the same type. Note that clusters containing only unlabelled vocalisations (detections with no expert annotation for its type) were not included since we cannot estimate if they are discriminant or not.

In addition to the estimated effort needed to find discriminant clusters, we report on the amount of annotations that would result from sorting these out (correcting on the potential 10% of wrongly clustered vocalisations). This is illustrated with the proportion of vocalisations that belong to discriminant clusters, in other words the proportion of the dataset that would be annotated using this algorithm. Eventually, the amount of labels that were not discriminantly clustered is also reported.

## Discussion

The results of this study suggest that deep representations self-supervisedly learnt more closely match experts' perception of vocalisation similarity than PAFs or whole spectrograms. This high level representation can be useful when it comes to vocal repertoire exploration and/or annotation, by using unsupervised clustering algorithms for instance. In general, efficiently annotating large amounts of vocalisations is useful for the development of automatic transcription systems, whether to directly train them or simply to assess their performances. This is especially true in the era of deep learning based bioacoustic classification systems [41, 52], a field for which one of the main limitation is the lack of annotated data for niche tasks such as non-human animal vocal transcription.

A significant challenge that comes along the use of deep-learning (whether in supervised or self-supervised settings) lies in hyper-parameters tuning and architecture design, in which there is a great degree of freedom and relatively few rationales to motivate choices. Nonetheless, given a fixed configuration (except for spectrogram settings which are driven by rationales), we show that our proposed method is relatively versatile: it is efficient in a wide range of species across taxa, despite diverse vocalisation frequency ranges and SNRs. Moreover, we demonstrate that the results are relatively robust to varying settings in input spectrogram normalisation, dimensionality reduction and clustering.

This versatility comes with one constraint however, spectrograms must come in dimensions that follow ($k_f 2^n \times k_t 2^n$). This is easily achieved by tuning the number of Mel filters and the

hop size accordingly, having a limited impact on the resulting spectrogram ($k$ can still vary to suit specific needs).

PAFs such as pitch estimates typically require expertise for settings to be tuned according to frequency ranges and SNR. Without such tuning, the yielded representation shows a relatively poor agreement with expert labels (Fig 5). Using whole spectrograms as proposed in previous studies [25, 26] mitigates this shortcoming, and yields more relevant representations. However, resulting performances can still be improved by using embeddings of an auto-encoder trained on spectrograms, as showed by an increase of 0.15 points of NMI for some species (Fig 5).

In addition to common handcrafted methods for vocalisation representation, we report on the use of generic deep audio embeddings for this task of vocalisation clustering. While representations of the HEAR challenge baseline show good performances when given to a classification head for a wide range of tasks [47], our experiments suggest that in most cases, without fine-tuning, they are less relevant than auto-encoders to measure vocalisation similarity described by expert labels. Nonetheless, results also suggest that an auto-encoder trained on a diverse set of vocalisations is useful to cluster a repertoire unseen in training. This strongly alleviates the effort needed to employ the auto-encoder approach on new repertoires: a 'generic' encoder can be used and no dataset specific training is required.

Learning vocalisation representations with auto-encoders has been studied in the past for zebra finch and mice [28], with an important difference made in the loss function used in training: a variational auto-encoder loss is used. We suggest to not include the gaussian constraint on the bottleneck distribution since it seems counter-productive for samples' clusterability. Rowe et al. [30] have also studied auto-encoders for bird species clustering. The authors employed a MSE loss and their approach does not outperform handcrafted features (MFCC). In this study, the use of the perceptual loss yields equal or better performing embeddings across datasets. Presumably, by focusing on the reconstruction error of highly salient spectrogram components, the perceptual loss reduces the amount of information related to background noise found in the bottleneck. Despite using the perceptual loss instead of a regular pixel loss, recordings sometimes contain background noises that create an unwanted variability in embeddings of vocalisations from the same type. Often, these background noises are relatively stationnary, and thus can be easily cancelled by substracting the median of each frequency bins of the spectrogram.

For the humpback whale complete dataset (as opposed to its 'small' counter part) the NMI between clusters and expert labels remains around 0.5. This might be explained by the graded nature of humpback whale unit types, which have been shown to evolve both within and across songs [53] (songs are are transmitted culturally, changing at every reproductive season [17]). To challenge this hypothesis, we ran the repertoire discovery algorithm on annotations of the secondary analyst of the study that provided the dataset [37], which gather 1,800 vocalisations recorded on a single day. The resulting agreement between clusters and expert labels rises around 0.7 of NMI, suggesting that the intra-type vocalisation variability (lowered when working with a limited time span) is responsible for poorer clustering results.

Vocalisation clustering is an important building block for the modelling of non-human animal communication systems. For this study, we chose to use discrete expert labels of vocalisation types as a proxy to measure the relevance of learnt representations. Being aware of the potential limitations of such ground truths, they were necessary to asses the applicability of the method until more data is collected specifically on animal perception [54]. We suggest that learnt representations and systematic clustering might be an opportunity to emancipate from the potential subjectivity of human labels, especially with soft cluster assignments helping with

graded repertoires and difficult boundary decisions [19] (it is also available for HDBSCAN [32]).

A potential shortcoming of this work is the omission of sequential context. Context has a significant impact on birds vocal perception [55], and was shown to improve the classification of humpback whale vocalisations [56]. We believe that jointly learning local categories with sequential organisation might be the next opportunity to advance unsupervised vocal sequence transcription, following the path led by speech applications [57].

## Conclusion

This paper proposes a framework to assist vocal repertoire discovery using deep representation learning with auto-encoders. It takes as input a set of detected vocalisations and automatically suggests categories by acoustic similarity. This demands only three compulsory manual settings that can be defined with very limited knowledge of vocalisation characteristics: the sample rate, the size of the Fourier transform window, and the signal duration suitable to describe each vocalisation. These settings are used to compute vocalisation spectrograms, which an auto-encoder is then trained to compress. The resulting high-level representation is projected with UMAP and clustered with HDBSCAN. This paper gives hints on how to choose settings for the two latter algorithms depending on dataset characteristics.

We experiment this auto-encoder framework on 8 different datasets with varying amounts of vocalisations, repertoire sizes, SNRs, and frequency ranges. Despite these variations, we demonstrate a relatively good agreement between the unsupervised categorisation and expert labels of vocalisation types, consistently superior to the baseline approach of using whole spectrograms. The two methods are compared both with a metric of information theory (NMI) and with indicators of the effort needed to manually annotate the dataset given clusters. Indeed, the method is not trustworthy for the direct transcription of vocal sequences, but can significantly reduce the human effort needed in characterising and annotating unknown vocal repertoires.

The procedure is made available as an open source Python package for the community to use it on animal vocal repertoire studies, including functionalities such as auto-encoder training, interactive browsing of embeddings, clustering, and plotting for efficient annotation. Also, we publish the pretrained weights of a 'generic' vocalisation encoder trained on all datasets of this paper, which is in most cases as good as a dataset specific auto-encoder in clustering vocalisations of a new repertoire.

## Supporting information

**S1 Fig. Impact of varying spectrogram frequency layouts and dynamic range compression for the subsequent vocalisation categorisation.**
(TIF)

## Acknowledgments

We are grateful to the researchers who shared their annotated datasets of humpback whale vocalisations (Franck Malige, Divna Djokic, and Renata Sousa-Lima). Humpback whale data collection was possible thanks to the logistic support from Projeto Baleia Jubarte sponsored by Petróleo Brasileiro S.A. (PETROBRAS). We are also grateful to the researchers that provided the dolphin signature whistles (Laela Sayigh and Vincent Janik). We acknowledge their significant contribution to this paper in this regard.

## Author Contributions

**Conceptualization:** Paul Best, Hervé Glotin, Ricard Marxer.

**Data curation:** Paul Best.

**Formal analysis:** Paul Best.

**Funding acquisition:** Hervé Glotin.

**Investigation:** Paul Best, Ricard Marxer.

**Methodology:** Paul Best, Ricard Marxer.

**Project administration:** Hervé Glotin.

**Resources:** Hervé Glotin.

**Software:** Paul Best.

**Supervision:** Sébastien Paris, Hervé Glotin, Ricard Marxer.

**Validation:** Hervé Glotin, Ricard Marxer.

**Visualization:** Paul Best.

**Writing – original draft:** Paul Best.

**Writing – review & editing:** Paul Best, Sébastien Paris, Ricard Marxer.

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
