## [Decision Letter · Decision Letter 0]

15 May 2023

PONE-D-23-06746Deep audio embeddings for vocalisation clusteringPLOS ONE

Dear Dr. Best,

Thank you for submitting your manuscript to PLOS ONE. After careful consideration, we feel that it has merit but does not fully meet PLOS ONE’s publication criteria as it currently stands. Therefore, we invite you to submit a revised version of the manuscript that addresses the points raised during the review process.

We look forward to receiving your revised manuscript.

Kind regards,

Jie Xie, Ph.D.

Academic Editor

PLOS ONE

Reviewers' comments:

Reviewer's Responses to Questions

**Comments to the Author**

1. Is the manuscript technically sound, and do the data support the conclusions?

Reviewer #1: Yes

Reviewer #2: Yes

2. Has the statistical analysis been performed appropriately and rigorously? 

Reviewer #1: Yes

Reviewer #2: Yes

3. Have the authors made all data underlying the findings in their manuscript fully available?

Reviewer #1: Yes

Reviewer #2: No

4. Is the manuscript presented in an intelligible fashion and written in standard English?

Reviewer #1: Yes

Reviewer #2: Yes

5. Review Comments to the Author

Reviewer #1: This paper therefore studies a new method for encoding vocalisations, allowing for automatic clustering to alleviate vocal repertoire characterisation. The authors use a convolutional auto-encoder network to learn an abstract representation of animal vocalisations. I believe the topic of this research is of interest to the related research community. Here are my detailed comments.

(1) Previous studies have used auto-encoders for studying ecoacoustic data, what’s the different between the current study and the following references?

Rowe B, Eichinski P, Zhang J, et al. Analyzing Big Environmental Audio with Frequency Preserving Autoencoders[C]//2021 IEEE 17th International Conference on eScience (eScience). IEEE, 2021: 70-79.

Rowe, Benjamin, et al. "Acoustic auto-encoders for biodiversity assessment." Ecological Informatics 62 (2021): 101237.

(2) In table 2, the authors use different parameter settings for different species and sources, how to set those parameters?

(3) One of the pitfalls of the STFT is that it has a fixed resolution. However, wavelet-based representation can solve this limitation. Why the authors choose STFT? A comparison between STFT- and wavelet-based representation is worth investigating.

Reviewer #2: The article proposes an innovative method for evaluating the quality of clusters formed by bioacoustic waveforms using an Autoencoder and UMAP. While the article is well written and generally easy to understand, there are a few points that could be improved.

- The discussion highlights the versatility of the proposed method compared to using pure spectrograms. However, it fails to address a significant challenge associated with the configuration of the Autoencoder's hyperparameters (layers, sizes, etc.). This challenge represents a disadvantage and an intrinsic difficulty of the proposed method, and it should be addressed and discussed in the article. In addition, it would be beneficial to include a more comprehensive analysis of the disadvantages or limitations of the proposed technique.

- In Table 3, the columns are difficult to understand, because for each column, with a unique heading, there are two subcolumns. 

- Towards the end of the discussion, it is mentioned that the method is versatile; however, it is stated that the recordings were downsampled to suited the method, rather than the other way around. This implies a lack of flexibility in the method's application due to the constraints imposed by the perceptual loss and VGG network, which requires square matrix inputs. It would be helpful to elaborate on this limitation and discuss potential workarounds or alternatives.

- The article should address the potential influence of background noise on bioacoustic systems and how it might affect cluster formation. High-capacity models like deep learning have the ability to learn features related to background noise, which could impact the clustering process. It is important to discuss this issue and conduct experiments to investigate whether such biases were avoided or mitigated in the proposed method.

- In Figure 3, the subplots are not convincing as there are many unclustered gray dots. The interpretation and inspection of these unclustered points should be clarified in the article. Additionally, the ideal UMAP and HDBSCAN parameters used in each graph should be provided, including any search process for determining the ideal UMAP parameters.

- Figure 4 shows that the NMI decreases for cassin_vireo as the UMAP dimension increases. The reason behind this trend should be explained in the article to provide a better understanding of the results.

- The sentence mentioning the graded nature of humpback whale unit types and their temporal instability needs further clarification. What does it mean for a signal to be "temporally unstable"? This concept should be elaborated upon to ensure readers fully comprehend its significance in the context of the article.

- The sentence discussing the increased agreement between clusters and expert labels (0.72 NMI) implies that additional experiments were conducted but not reported. These experiments and their findings should be included in the article to provide a complete and transparent account of the research. Furthermore, the phrase "reduced when working with a limited amount of time" requires clarification regarding the specific time limit or duration used for normalization of the AE inputs.

- The GitHub repository should be better organized and documented more effectively. A detailed tutorial on replicating the experiments and processing new datasets should be added to facilitate reproducibility and enable others to build upon the research.

- The work at the provided link (https://experiments.withgoogle.com/bird-sounds) is very nice. It would be interesting to know if the authors have plans to develop a web server for their application, as it could be highly beneficial for the community. (suggestion)

6. PLOS authors have the option to publish the peer review history of their article (what does this mean?). If published, this will include your full peer review and any attached files.

Reviewer #1: No

Reviewer #2: No

---

## [Author Response · Author response to Decision Letter 0]

21 Jun 2023

S1_Fig1.tif has been renamed to S1_Fig.tif and the asterisk for the corresponding author has been added

Auto-encoder weights, vocalization embeddings, and ground truth labels of vocalization types that are necessary to produce performance metrics and plots in this study are available on a figshare repository : https://doi.org/10.6084/m9.figshare.23138210.v1

The acoustic data that were used to train auto-encoders and generate embeddings along with their expert label are not owned by the authors but might be accessed via their respective sources :

bengalese finch1 : 

Nicholson, David; Queen, Jonah E.; J. Sober, Samuel (2017): Bengalese Finch song repository. figshare. Dataset. https://doi.org/10.6084/m9.figshare.4805749.v5

bengalese finch2 :

Koumura, Takuya (2016): BirdsongRecognition. figshare. Media. https://doi.org/10.6084/m9.figshare.3470165.v1

cassin vireo, california thrasher, black-headed grosbeak :

Arriaga, J. G., Cody, M. L., Vallejo, E. E., & Taylor, C. E. (2015). Bird-DB: A database for annotated bird song sequences. Ecological Informatics, 27, 21-25.

https://taylor0.biology.ucla.edu/birdDBQuery/

humpback whale and humpback whale (small) (upon request) :

Malige, F., Djokic, D., Patris, J., Sousa-Lima, R., & Glotin, H. (2021). Use of recurrence plots for identification and extraction of patterns in humpback whale song recordings. Bioacoustics, 30(6), 680-695. (contact : Franck Malige, franck.malige@lis-lab.fr)

Dolphin (upon request) :

Sayigh L, Janik VM, Jensen F, Scott MD, Tyack PL, Wells R. The Sarasota dolphin whistle database: A unique long-term resource for understanding dolphin communication. Frontiers in Marine Science. 2022;. (contact Laela S. Sayigh, lsayigh@whoi.edu)

The preprint “Sainburg T, Thielk M, Gentner TQ. Latent space visualization, characterization, and generation of diverse vocal communication signals; 2020” was referenced rather than its peer-reviewed version because the first mentions experiments with auto-encoders but the latter doesn’t. This reference has been removed.

Reviewer #1:

This paper therefore studies a new method for encoding vocalisations, allowing for automatic clustering to alleviate vocal repertoire characterisation. The authors use a convolutional auto-encoder network to learn an abstract representation of animal vocalisations. I believe the topic of this research is of interest to the related research community. Here are my detailed comments.

(1) Previous studies have used auto-encoders for studying ecoacoustic data, what’s the different between the current study and the following references?

Rowe B, Eichinski P, Zhang J, et al. Analyzing Big Environmental Audio with Frequency Preserving Autoencoders[C]//2021 IEEE 17th International Conference on eScience (eScience). IEEE, 2021: 70-79. Rowe, Benjamin, et al. "Acoustic auto-encoders for biodiversity assessment." Ecological Informatics 62 (2021): 101237.

These references report on clustering vocalizations by species, using 100 randomly sampled signals (10 from 10 species), and the proposed auto-encoder does not outperform handcrafted features (MFCC). Our study clusters vocalizations by types of a repertoire within a species, uses larger datasets (some repertoires gather a 100 classes), and our proposed architecture systematically outperforms handcrafted features.

(2) In table 2, the authors use different parameter settings for different species and sources, how to set those parameters?

The section “Signal preprocessing and Fourier transform” in Materials and Methods details how those parameters were set and gives rational for a user to set them for a new specie

(3) One of the pitfalls of the STFT is that it has a fixed resolution. However, wavelet-based representation can solve this limitation. Why the authors choose STFT? A comparison between STFT- and wavelet-based representation is worth investigating.

This paper intends to propose a method that is fast, easily usable by users that are not experts in signal processing, and applied to non-transient harmonic signals. This makes the Fourier transform a good a priori representation for them. We’ve added a justification for this in the “signal processing and Fourier transform” section in Material and Methods.

Reviewer #2: The article proposes an innovative method for evaluating the quality of clusters formed by bioacoustic waveforms using an Autoencoder and UMAP. While the article is well written and generally easy to understand, there are a few points that could be improved.

- The discussion highlights the versatility of the proposed method compared to using pure spectrograms. However, it fails to address a significant challenge associated with the configuration of the Autoencoder's hyperparameters (layers, sizes, etc.). This challenge represents a disadvantage and an intrinsic difficulty of the proposed method, and it should be addressed and discussed in the article. In addition, it would be beneficial to include a more comprehensive analysis of the disadvantages or limitations of the proposed technique.

Experiments demonstrate that there is no need for additional hyperparameter configuration for the method to be efficient on a wide variety of signals. We’ve added a statement in the discussion’s second paragraph that notes this challenge that comes with deep learning.

- In Table 3, the columns are difficult to understand, because for each column, with a unique heading, there are two subcolumns.

This Table has been updated with sub-headings for the sub-columns.

- Towards the end of the discussion, it is mentioned that the method is versatile; however, it is stated that the recordings were downsampled to suited the method, rather than the other way around. This implies a lack of flexibility in the method's application due to the constraints imposed by the perceptual loss and VGG network, which requires square matrix inputs. It would be helpful to elaborate on this limitation and discuss potential workarounds or alternatives.

For humpback whales, recordings were downsampled only to yield more relevant spectrograms (avoiding to have half of the image above their highest vocalizations). We believe this does not imply a lack of flexibility of the method. Some constraints do occur for the input of the encoder, but having square spectrograms is not one of them. We’ve added a sentence in the discussion in this regards : “this versatility comes with one constraint…”.

- The article should address the potential influence of background noise on bioacoustic systems and how it might affect cluster formation. High-capacity models like deep learning have the ability to learn features related to background noise, which could impact the clustering process. It is important to discuss this issue and conduct experiments to investigate whether such biases were avoided or mitigated in the proposed method.

A sentence has been added to the discussion regarding the question of background noise, giving hints on mitigation methods.

- In Figure 3, the subplots are not convincing as there are many unclustered gray dots. The interpretation and inspection of these unclustered points should be clarified in the article. Additionally, the ideal UMAP and HDBSCAN parameters used in each graph should be provided, including any search process for determining the ideal UMAP parameters.

As described in the legend of Fig 3, colors denote labels, and not clusters (this has been clarified in the legend). Also, the result section describes “we used a grid search to tune these parameters…”; “...these generic settings were kept for all experiments”.

- Figure 4 shows that the NMI decreases for cassin_vireo as the UMAP dimension increases. The reason behind this trend should be explained in the article to provide a better understanding of the results.

There was a mismatch with legends and plots in this figure, which has been corrected? Also, a small decrease in performance with the increasing number of dimensions is hardly interpretable, as there is no consistent trend across datasets. Unfortunately, the authors are only able to provide empirical evidence here.

- The sentence mentioning the graded nature of humpback whale unit types and their temporal instability needs further clarification. What does it mean for a signal to be "temporally unstable"? This concept should be elaborated upon to ensure readers fully comprehend its significance in the context of the article.

This sentence has been changed to be clarified

- The sentence discussing the increased agreement between clusters and expert labels (0.72 NMI) implies that additional experiments were conducted but not reported. These experiments and their findings should be included in the article to provide a complete and transparent account of the research. Furthermore, the phrase "reduced when working with a limited amount of time" requires clarification regarding the specific time limit or duration used for normalization of the AE inputs.

The “humpback whale (small)” dataset has been added to the paper along with the results it yields, reporting all experiments that are mentioned in the text. The time span referred to in “working with a limited time span” is made clear by the preceding sentence (“in a single day”).

- The GitHub repository should be better organized and documented more effectively. A detailed tutorial on replicating the experiments and processing new datasets should be added to facilitate reproducibility and enable others to build upon the research.

The repository has been better organized and documented in this regard.

- The work at the provided link (https://experiments.withgoogle.com/bird-sounds) is very nice. It would be interesting to know if the authors have plans to develop a web server for their application, as it could be highly beneficial for the community. (suggestion)

Such interface demands a significant engineering effort for which authors of this study unfortunately do not have time for. It is possible for other contributors to use our proposed methods and embeddings for such an interface however.

---

## [Editor Report · Decision Letter 1]

26 Jun 2023

Deep audio embeddings for vocalisation clustering

PONE-D-23-06746R1

Dear Dr. Best,

We’re pleased to inform you that your manuscript has been judged scientifically suitable for publication and will be formally accepted for publication once it meets all outstanding technical requirements.

Kind regards,

Jie Xie, Ph.D.

Academic Editor

PLOS ONE
---

## [Editor Report · Acceptance letter]

29 Jun 2023

PONE-D-23-06746R1 

Deep audio embeddings for vocalisation clustering 

Dear Dr. Best:

I'm pleased to inform you that your manuscript has been deemed suitable for publication in PLOS ONE. Congratulations! Your manuscript is now with our production department. 

Kind regards, 

on behalf of

Dr. Jie Xie 

Academic Editor

PLOS ONE